# RORγ Structural Plasticity and Druggability

**DOI:** 10.3390/ijms21155329

**Published:** 2020-07-27

**Authors:** Mian Huang, Shelby Bolin, Hannah Miller, Ho Leung Ng

**Affiliations:** 1Department of Biochemistry and Molecular Biophysics, Kansas State University, Manhattan, KS 66506, USA; mianhuang@ksu.edu (M.H.); hannahm59@ksu.edu (H.M.); 2Division of Biology, Kansas State University, Manhattan, KS 66506, USA; shelbybolin@ksu.edu

**Keywords:** RORγ, plasticity, druggability, orthosteric binding pocket, allosteric binding pocket

## Abstract

Retinoic acid receptor-related orphan receptor γ (RORγ) is a transcription factor regulating the expression of the pro-inflammatory cytokine IL-17 in human T helper 17 (Th17) cells. Activating RORγ can induce multiple IL-17-mediated autoimmune diseases but may also be useful for anticancer therapy. Its deep immunological functions make RORɣ an attractive drug target. Over 100 crystal structures have been published describing atomic interactions between RORɣ and agonists and inverse agonists. In this review, we focus on the role of dynamic properties and plasticity of the RORɣ orthosteric and allosteric binding sites by examining structural information from crystal structures and simulated models. We discuss the possible influences of allosteric ligands on the orthosteric binding site. We find that high structural plasticity favors the druggability of RORɣ, especially for allosteric ligands.

## 1. Introduction

Retinoid-related orphan receptor γ (RORγ) is a member of the nuclear receptor (NR) superfamily in the human genome, with two isoforms, the thymus-specific RORγt and the ubiquitous RORγ. The first isoform plays vital roles in promoting T cell differentiation into the T helper 17 (Th17) subtype and regulating IL-17A gene transcription in Th17 cells [1]. The second isoform is believed to act as a negative regulator of adipocyte differentiation to impact adipogenesis and insulin sensitivity [2]. It may also be involved in supporting the Th17 differentiation [3]. Overexpression of RORγ and RORγt can lead to obesity-associated insulin resistance and multiple autoimmune diseases such as multiple sclerosis, rheumatoid arthritis, Crohn’s disease and psoriasis. On the other hand, high RORγt activity can sometimes be beneficial and has been shown to enhance antitumor immunity by increasing IL-17, GM-CSF (granulocyte-macrophage colony-stimulating factor) secretion and decreasing PD-1 expression [4,5]. The pathological functions and the therapeutic potential of RORγ make it an intriguing drug target. Besides that, several diseases are also associated with Th17 cells. For nonalcoholic fatty liver disease, increased hepatic Th17 cells, RORγt and IL-17 expression are observed in the progression from one subtype nonalcoholic fatty liver disease to the other subtype nonalcoholic steatohepatitis [6,7]. Also, Th17 cells and IL-17 play mixed pro-tumor roles in the development of colorectal cancer [8]. Given the regulatory effects of RORγ isoforms in Th17 differentiation and IL-17 transcription, discovering drugs targeting RORγ may provide alternative therapeutic strategies for these Th17-mediated diseases.

The RORγ isoforms only differ in the first 21 residues, which are truncated in RORγt. To simplify the discussion in this article, we use the term “RORγ” to represent both isoforms and number the residues by following the ubiquitous isoform. Similar to other NRs, RORγ is composed of an N-terminal domain (NTD, 1–30 a.a.), a DNA-binding domain (DBD, 31–95 a.a.), a hinge domain (HD, 96–265 a.a.), a ligand-binding domain (LBD, 266–505 a.a.) and a C-terminal domain (CTD, 506–518 a.a.). The knowledge obtained from other NRs facilitates scientists to quickly recognize the orthosteric binding site located in the LBD. Under the engagement of pharmaceutical companies, including Merck, Takeda and Biogen, in developing novel RORγ-targeting drugs, hundreds of small-molecule modulators binding in the orthosteric site have been developed and structurally investigated. However, numerous highly active compounds were discontinued or suspended for further development in the stages of preclinical studies or clinical trials, owing to off-target effect, toxicity and poor therapeutic efficacy [9,10]. It is disappointing that no compound has yet passed Phase II clinical trials. More recently, two allosteric binding sites have been discovered in the LBD [11] and HD [12] and relative inverse agonists were developed. In contrast to the orthosteric site conserved in the NRs, these allosteric sites exist in unique binding pockets different from other NRs, resulting in better binding selectivity. Also, the non-overlapping locations of the sites avoid competition between allosteric inverse agonists and endogenous agonists, leading to lower dose usage in treatments and potentially developing fewer side effects [13]. Therefore, exploring allosteric RORγ modulators is a hot research topic now.

The flexibility of a protein is necessary for accomplishing its function [14,15,16]. NRs are multidomain proteins and activation of the transcriptional function is achieved through a signal initiated by an agonist ligand binding in the orthosteric site and then transferred from LBD to DBD under the mediation of HD and coactivator proteins. The whole process involves multiple steps of intra- and inter-domain actions accompanied by conformational changes. Studying these changes will not only shed light on the mechanisms of protein activity but also aid in identifying new allosteric binding sites for drug discovery. 

No full-length three-dimensional experimental structure of RORγ is currently available. The LBD is the only domain that has been successfully co-crystallized with ligands. Over 100 crystal structures of RORγ LBD-ligand complexes are published in the Protein Data Bank (PDB), including 69 orthosteric inverse agonists, 18 orthosteric agonists and 9 allosteric inverse agonists. Many of the released modulators have been classified by chemotype and described in a recent review [17]. Here, complementarily, we focus on RORγ itself and use the crystal structures as snapshots to glimpse its structural plasticity. We discuss the relationship between druggability and plasticity of RORγ by covering its orthosteric and allosteric binding pockets. We also summarize computational and modeling results that provide more insight into structural dynamics and its role in druggability.

## 2. The Plasticity of the RORγ Orthosteric Binding Site Affects Druggability

The orthosteric binding site of RORγ is in the core of the globular LBD and the pocket is mainly embraced by α-helices H3, H5, H6, H7 and H11 with the critical H12 in the vicinity of one pocket exit (Figure 1). Since the putative endogenous agonists, cholesterol analogs (hydroxycholesterols [18], 4α-carboxy and 4β-methyl-zymosterol [19]), were found binding in this site; thousands of small molecules have been screened as potential ligands and the structure-activity relationships (SARs) have been thoroughly studied. It is believed that a RORγ agonist binding in the orthosteric site facilitates contact of the coactivator with the LBD surface, initiating the transcriptional function. In contrast, a RORγ inverse agonist in the same site achieves inhibition by preventing the coactivator binding to the LBD. Depending on the ligand type, H12 changes its conformation to act as an on-and-off switch. Meanwhile, the shape of the pocket is plastic and can demonstrate induced-fit binding.

### 2.1. The Dynamics of H12 

For a long time, H12 was primarily considered as a static helix distal from the orthosteric pocket in apo NR LBDs, based on artificial crystal-packing interactions in co-crystals of the RARγ LBD-ligand complexes [20]. This changed in 2003 when Kallenberger et al. proposed a more dynamic model based on tracking the H12 movement in the PPARγ LBD by fluorescence anisotropy techniques [21]. They found that H12 is highly dynamic and disordered in the apo LBD but agonist binding induces H12 to become a helix and lie on the pocket exit with a well-defined pose, facilitating recruitment of the coactivator to the LBD surface and consequently activating transcription. 

This dynamic model also applies to the H12 behavior in the RORγ LBD. NMR results showed that the H12 chemical shifts in the unbound LBD were missing or shifted in backbone ^15^N-TROSY and methyl-^13^C-HSQC spectra, demonstrating disorder [22]. This was reflected in the difficulty of crystallizing the apo LBD in the absence of the coactivator [22,23]. HDX-MS results also demonstrated reduced dynamic properties of H12 when the RORγ LBD transitioned from the apo state to the agonistic state, supporting a disorder-to-order transition [24]. However, the H12 dynamic properties in the apo and inverse-agonist states are similar, suggesting that H12 may not have a discrete conformation when an inverse agonist binds in the orthosteric site. In some co-crystals of RORγ LBD with inverse agonists, H12 could not be modeled due to missing electron density [25]; in many cases, H12 had to be truncated to cocrystallize the RORγ LBD with inverse agonists [26,27,28]. 

The term “agonist lock” is introduced to describe the atomic mechanism of LBD agonism [26]. In the agonist-bound LBD, a hydrogen bond formed between His^479^ and Tyr^502^ locks H12 in the correct position (Figure 2), facilitating coactivator recruitment. Meanwhile, Phe^506^ assists in stabilizing the lock by π-stacking with His^479^ [22]. It is now commonly accepted that RORγ orthosteric binders mainly regulate the transcriptional function by manipulating the stability of the lock. The results of molecular dynamics (MD) simulations further confirm it by showing lower fluctuations in interaction energy for the lock in the agonist-bound LBD and higher fluctuations for the lock in the unbound or inverse agonist-bound LBD [29]. Furthermore, they provide evidence for the existence of the lock in the unbound state, even if energetically unfavorable, implying that the H12 conformation in the unbound LBD crystal could be one of many conformations switching in solution. 

### 2.2. The Plasticity of the Orthosteric Binding Pocket on Accepting Modulators

The pocket in apo RORγ features a druggable volume of 940 Å^3^ with 61% hydrophobicity [22]. The agonists co-crystallized with the LBD only partially occupy it (Figure 3). In contrast, the size and molecular weight features of RORγ inverse agonists deviate more than the agonists, with agonists having a maximum molecular weight <600 g/mol. To decipher the SAR rules, we take a closer look at the conformational changes of the orthosteric site induced by modulators. 

Among 18 co-crystals of LBD-agonist complexes (Appendix A), all LBDs share similar conformations with backbone C root-mean-square deviations (RMSDs) lower than 1.2 Å. The binding modes of the agonists are similar, even for the 12 which have diverse non-sterol scaffolds. The modulators interact non-covalently with H3, H5, H6, and/or H7 to stabilize the whole pocket and hydrophobically contact His^479^ while remaining distant from Tyr^502^ (Figure 4). Both the pocket and the agonist lock are also observed in the ligand-less state bound to a coactivator peptide. It suggests a rigid, uniform model of RORγ LBD required to achieve activation, which must be considered in the structure-based drug discovery of RORγ agonists. To date, the known agonists were found accidentally or were synthesized based on the scaffolds of inverse agonists. Virtual screening (VS) may be helpful for the discovery of diverse agonist chemotypes. We will discuss the applications of VS in a later section. 

We also compared the conformations of orthosteric pockets in 69 LBD-inverse agonist complexes from crystal structures (Appendix A). Based on the pairwise backbone C RMSDs, we assigned them into three pocket conformation modes (Figure 5). Most of them are in Mode I with RMSDs <2 Å, implying a stable pocket structure. The Mode I pocket shows minor global difference to the apo pocket but we can still observe subtle conformational changes induced by modulators. The inverse agonists in this group pushed or pulled H3, H4, H5, H7 and/or H11 to slightly deviate from the original positions (Figure 6a). The side chain of His^479^ may flip (Figure 6d), which breaks the agonist lock [31]. Even if the lock is retained and the coactivator is recruited, the shifts of the helices moderately change the contacts to the coactivator (Appendix A) [32,33]. More investigation is needed to interpret the possible mechanisms of the inverse agonism caused by the subtle adjustment. In the Mode II models (RMSDs: 2–3 Å), the movement of the related helices, especially H11, becomes more evident (Figure 6b) and the side chain of His^479^ points to the direction opposite to the agonist lock (Figure 6e), leading to a broken lock [23,34]. For the Mode III pocket (RMSDs: 3–5 Å), in addition to the displaced helices alike to Mode II, part of H11 turns into a loop, where His^479^ is situated (Figure 6c,f). This dramatic change impairs the lock [35,36].

These modes construct a noticeably plastic structure of the orthosteric pocket for accepting inverse agonists. The flexibility is contributed from the limited shifts of H3, H4, H5 and H7 and the more dramatic changes of H11, H11,’ H12. Because the helices contact each other, a moderate movement of one helix may cause a large scale shift. Most inverse agonists not only form hydrogen-bonds to or hydrophobically contact His^479^ to affect the agonist lock but also indirectly affect the more displaceable helices through the more rigid helices, resulting in further pushing Tyr^502^ away from His^479^ and complete dysfunction of the lock. Therefore, the pocket plasticity offers several discrete ways for inverse agonists to block protein activation and can accommodate a broad range of inverse agonist chemotypes. This suggests that discovering inverse agonists should be easier than agonists, which has turned out to be historically accurate.

## 3. The Plasticity of RORγ Allosteric Binding Sites Increases Druggability

Drug companies have made extensive efforts to develop drugs targeting the RORγ orthosteric binding site for treating autoimmune diseases and cancers but numerous challenges have arisen in cell-based assays, animal experiments and clinical trials. A serious problem is off-target effects. Despite that the agonist lock is unique to the ROR subfamily, the orthosteric binding site is structurally conserved in most NRs. Some modulators under development have been found to tightly bind other NRs. While chemists are trying to optimize the structures to improve selectivity, other researchers are transferring attention to investigate the druggability of allosteric binding sites. 

### 3.1. The AF-2 Allosteric Binding Site 

Helix H12 in the RORγ LBD is also named activation function-2 (AF-2) for its role in interacting with cofactors and regulating activation. In 2015, Scheepstra et al. identified a novel allosteric binding site on AF-2 when investigating the SARs of a series of RORγ inverse agonists (**1**, **2**, **3**, **4**) (Figure 7) [11]. This site is a shallow pocket nested by H3, H4, H5, H11, lidded by H11′ and H12 and mainly filled with hydrophobic residues (Figure 8a,b). Ligands fit in the pocket through π-π stacking, hydrophobic effects and several hydrogen bonds. They force H12 to relocate, which dramatically extends the distance between His^479^ and Tyr^502^, for example, from 2.8 Å to 16.8 Å and interrupts the agonist lock (Figure 8c). Recently, more analogs (**5**, **6**) of **1** and modulators with different scaffolds targeting the site (**7**, **8**, **9**) have been developed. Interestingly, all allosteric inverse agonists directly hydrogen-bond to Ala^497^ and/or Phe^498^ on H11′ with or without one-to-two more bonds to the residues on H3 and H4. Ala^497^ is not conserved in the ROR family, making it a potential interacting target for improving drug selectivity. There may be other ligand binding modes available in this pocket still waiting to be discovered. 

We compared the binding modes of nine ligands (Figure 7). Compounds **1–8** were co-crystallized with the LBD [11,37,38,39,40]. Compound **9** was modeled from MD simulations [41]. The eight co-crystal structures overlap with the backbone RMSDs <1.5 Å, supporting the same conformation of the AF-2 allosteric pocket that occurs in all the structures. By comparing the volumes of the allosteric inverse agonists with the orthosteric ones, the pocket appears to tolerate only a narrow range of molecules. However, we believe the pocket can adjust to accept a broader range of ligands for two reasons. First, the pocket is surprisingly pliable. It is almost completely closed (<1 Å^3^) in the unbound state until a ligand induces expansion. The structural difference between the unbound and the bound LBDs is 2.7–3.0 Å in backbone RMSDs, demonstrating a notable conformational change before and after ligand binding. Specifically, induced by **1**, for example, helices H3, H4 and H11 shift towards the pocket center, H11′ loses the helical shape and H12 is pushed to the environment by **1** (Figure 8a,b). Second, the AF-2 pocket is connected to the LBD orthosteric pocket through a short tunnel, that can accommodate parts of some ligands (Figure 8d). The AF-2 pocket is also connected to the solvent environment by gaps between the helices, which allows ligands to protrude from the pocket. Thus, the pocket is extendable depending on the ligand chemistry.

For allosteric inverse agonists that bind in the AF-2 pocket, the mechanism for inhibiting transcriptional activation by RORγ remains to break the agonist lock by disrupting the interaction between His^479^ and Tyr^502^. Like the orthosteric pocket, the flexibility of the AF-2 pocket mainly comes from the dynamic properties of H11, H11′ and H12. However, some characteristics make the AF-2 pocket superior for drug targeting. In contrast to the buried orthosteric pocket, the AF-2 pocket is located on the surface of the LBD, which makes it directly accessible to ligands. The dynamic H11′ and H12 helices act as a mobile forearm and hand to hold the ligand. What is more, H11′ is more accessible for a ligand sitting in this pocket than the orthosteric one. H11′ is another unique feature only existing in the ROR family and is not conserved among RORα, RORβ and RORγ [22,41]. Targeting the residues in H11′ provides opportunities for increased drug selectivity.

### 3.2. An Allosteric Binding Site in the RORγ HD 

Previous research on NRs has revealed the vital role of the HD in the regulation of DBD-DNA interactions via posttranslational modifications [42,43,44]. It is not surprising that mutating Ser^113^ and Leu^114^ in the HD abolished ubiquitination at Lys^90^, which is required for TH17 differentiation ^41^. Based on this discovery, Lao et al. performed high-throughput screening (HTS) targeting the LBD and HD [12]. The SAR of the most active molecule (**10**) from the screen was explored through docking against a homology model of the full-length RORγ structure created using I-TASSER software [45]. The docking results predicted binding in an allosteric pocket in the HD between the DBD and LBD, where **10** hydrogen-bonds to Gln^223^ and hydrophobically contacts Leu^244^. Unfortunately, no crystallography was performed to validate the predicted binding mode.

To the best of our knowledge, this is the first report of an allosteric binding site in an NR HD. The paper only provides limited structural data on describing the allosteric pocket and the atomic mechanism of the inhibition. Thus, to understand the interactions of **10** with Ser^113^ and Leu^114^, we tried to reproduce the reported complex through docking **10** into a full-length RORγ model also produced from I-TASSER (Figure 9a, purple, Appendix A). One of the top-ranked binding poses occupies a position close to Lao’s but interacts with the surrounding residues differently (Figure 9b). The carbonyl oxygen of **10**, instead of the carboxyl oxygen as shown in the paper, hydrogen-bonds closely to Gln^223^. Leu^244^ is too far to interact with **10**. In our protein model, Ser^113^ and Leu^114^ are located at the edge of the pocket. We propose that the ligand occupies the adjacent space and prevent the ubiquitination-related enzymes from accessing these two residues. Meanwhile, we found that the structure of the modeled LBD deviated significantly from the crystal structures (Cα RMSD = 5.1 Å; Appendix A), which would adversely affect the performance of docking. As a control experiment, we also docked **10** into inverse-agonist-bound LBD crystal structures and found it was predicted to bind in the orthosteric site (Figure 9f, Appendix A). We conclude that docking alone cannot decide the correct binding site for **10**. Further experimental research is needed to confirm whether it binds in the originally proposed allosteric site.

Despite the unresolved question of the proper binding site for **10**, the study by Lao et al. is important for raising the possibility of new allosteric binding pockets in RORγ and other NRs. As a linker between the DBD and LBD, the HD is a flexible loop that may easily reshape itself to accommodate the chemistry of a ligand. Aside from the simulated model used above, I-TASSER also produced two other top-ranked models (Figure 9a, tan and salmon) with HDs adopting different conformations (Cα RMSDs: 7.3–13.0 Å; Appendix A). **10** can be docked at different sites in the DBD-LBD gap of two models (Figure 9c–e). In the absence of a crystal structure, these simulated protein models represent potential conformations of the whole RORγ, which may or may not correspond to reality. The docking results may not provide answers to the binding mode of **10** but do demonstrate a high degree of plasticity of a potential allosteric pocket in the HD. Additionally, the HD is the least conserved domain in NRs (even in the ROR subfamily). Hence, drugs targeting an allosteric site in the RORγ HD may provide better selectivity.

### 3.3. Other Potential Allosteric Binding Sites 

It is generally true that every member of the NR superfamily is activated by an agonistic signal transferred from the LBD to the DBD, mediated by a coactivator protein and the HD. Interrupting any step in this process can block activation.

In the agonist-bound conformation of RORγ, the coactivator peptide binds to a shallow, hydrophobic pocket formed by H3, H4 and H12 of the LBD (Figure 10a). Its small surface area makes it a challenging site for drug design. Also, the pocket is sequentially and structurally conserved in the ROR family, suggesting low selectivity as a drug target site. However, it still can be useful in drug targeting by joining it to the neighboring AF-2 allosteric site (Figure 10b). This extended pocket introduces more contact area to ligands, providing opportunities to improve binding affinity.

In each NR, the DBD contains two zinc finger motifs that recognize and bind to a specific DNA sequence, called the hormone response element (HRE). Recently, Ban et al. reported developing an allosteric inverse agonist targeting the DBD of the androgen receptor by taking advantage of a pocket visible in the DBD-HRE complex crystal structure [46]. This raises the fascinating possibility of identifying allosteric DBD binding pockets in other NRs. Several questions need to be answered before attempting this for RORγ. It is known that mutations at Ser^38^, Lys^52 and^ Lys^90^ decrease the binding of the RORγ DBD to the HRE [47,48,49]. The residues are situated in the zinc finger motif, SUMOylation site and ubiquitination site of the DBD, respectively, which are potential candidate sites for drug targeting. Unfortunately, no co-crystal structure of the RORγ DBD-HRE complex is available to explore the druggability of these sites. A more pressing issue is that the high conservation of the DBD (especially for these three residues) in the ROR subfamily reduces the opportunity of designing selective drugs.

## 4. The Helices Acting as the Arm and Hand of the LBD 

Two ligand-binding sites have been identified in the RORγ LBD, including the orthosteric site and the AF-2 allosteric site. Interestingly, the plasticity of both pockets is determined by the dynamic properties of helices H11, H11′ and H12 as induced by modulators (Figure 11). When an agonist binds in the orthosteric site, the helices fix it in the pocket through forming a rigid triangular net constructed of hydrogen bonds with or without water mediation, hydrophobic packing and π stacking effects. When an inverse agonist binds in either site, the triangular net is broken and the helices display diverse degrees of freedom. Anchored by H10 in one end and restricted by the other helices, H11 can only move in a limited range of angles, serving as an arm in the LBD. In contrast, H11′ and H12 are as mobile as a free hand connecting to a moving arm. Because it is the C-terminal end of the protein, H12 is presumably more active than H11’. Alternatively, H12 may be bound to the LBD surface by interacting with either ligand and/or residues. Researchers must also be cautious about interpreting the dynamic properties of H12 due to artefactual crystal packing interactions in crystal structures. Nevertheless, H11′ is found disordered in all co-crystals bound with allosteric inverse agonists and most of the co-crystals with orthosteric inverse agonists, whether H12 is in the helical conformation or not. H11′ heavily relies on hydrogen bonding from H11 and H12 to retain it in the helical conformation, so disruption from modulators can release it from these restraints. Monitoring the conformational change of H11′ may help to predict modulator activity. 

## 5. The Role of Protein Plasticity in the Computer-Aided Discovery of RORγ-Targeting Drugs

In contrast to the classical HTS method of screening chemicals for RORγ-targeting drugs, VS accomplishes the searching process in silico by docking a library of drug-like small compounds into a structural model, scoring each compound by mathematic algorithms. It immensely saves time and cost in the initial screening stages of narrowing down the chemical pool. In 2014, the Xu group conducted VS on a 220,000-compound database against the orthosteric pocket in an inverse-agonistic LBD crystal model using the Schrödinger Glide software, resulting in 115 hits [50]. In 2016, they did another VS to a five-times larger database against the same model by using Schrödinger Glide and Phase simultaneously, resulting in 28 hits [51]. Most of the hits showed moderate or strong activity in biological and biochemical assays but the models of the atomic structures for the best hits were only modeled. The pharmacophore of an allosteric inverse agonist (**8**) bound in the AF-2 site was also discovered by VS against the crystal model induced by **1** and its SAR was thoroughly studied by X-ray crystallography [40]. These illustrate the feasibility of applying VS in the discovery of RORγ LBD-targeting modulators. It is important to note that a feature of VS is that the programs usually dock the compounds against a rigid model to speed up the screening process. It seems indifferent to the search of new agonists since the crystal models before and after ligand binding are almost identical. However, some potential inverse agonists could be omitted by the program if the docked model is remarkably different from the real shape of the bound pocket. Therefore, the plasticity of the pocket needs to be considered in VS to avoid missing some important hits. The state-of-the-art docking methods incorporate multiple model templates and machine learning scoring functions to markedly improve the effectiveness of predicting ligand activity and even the quantitative predictions of binding affinity [52].

Another in silico method, MD, has aided the study of the H11, H11,’ H12 flexibility affected by different types of ligands and the atomic mechanisms on activating or inactivating RORγ and other NRs [29,53,54]. It is also an alternative method to study the SAR for a protein-ligand complex that lacks a co-crystal structure. For example, an MD simulation on the complex of **10** and the full-length RORγ homology model (Figure 9b) could provide a view on the possible atomic interactions, the induced conformational change of the protein and the stability of the complex, because the ligand may drop from the pocket during the simulation if the binding is not strong enough. 

## 6. Conclusions

The RORγ gene was first discovered in 1994 [55] and its role in immune homeostasis was unclear until 2006 [1]. Although enormous progress has been made to understand it structurally and biologically since that time, much mystery is still waiting to be uncovered. In this review, we highlighted the role of the protein structural plasticity in ligand binding and activity and drug design. RORγ is highly plastic and can form multiple modulator binding pockets. Consequently, it adopts diverse conformations and fulfills different duties. This characteristic makes RORγ a superb drug target for treating autoimmune diseases and cancers. X-ray crystallography is still the gold standard technique in studying atomic protein-ligand interactions but its strict requirement on protein stability makes crystallization of the full-length RORγ unreachable for now. Fortunately, other methods such as homology modeling and cryo-electron microscopy are developing to compensate for the deficiency. Other in silico methods, such as MD and VS, also play increasingly important roles in investigating the dynamics and drug screening of RORγ. Under a combination of experimental and computational methods, we expect that the understanding of RORγ will be substantially improved in the near future and more RORγ-targeting drugs will be developed with better clinical performance.

## Figures and Tables

**Figure 1 ijms-21-05329-f001:**
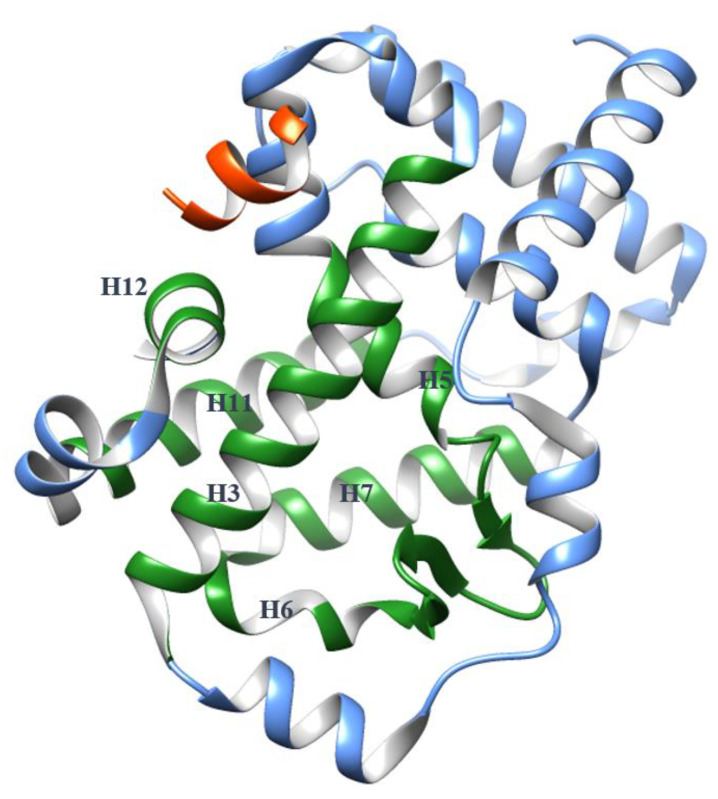
The globular structure of the apo RORγ ligand-binding domain (LBD) (Protein Data Bank (PDB) 5X8U) with a coactivator (orange). H3, H5, H6, H7, H11 and H12 are presented in green color to indicate the orthosteric binding pocket, while other helices are presented in blue color.

**Figure 2 ijms-21-05329-f002:**
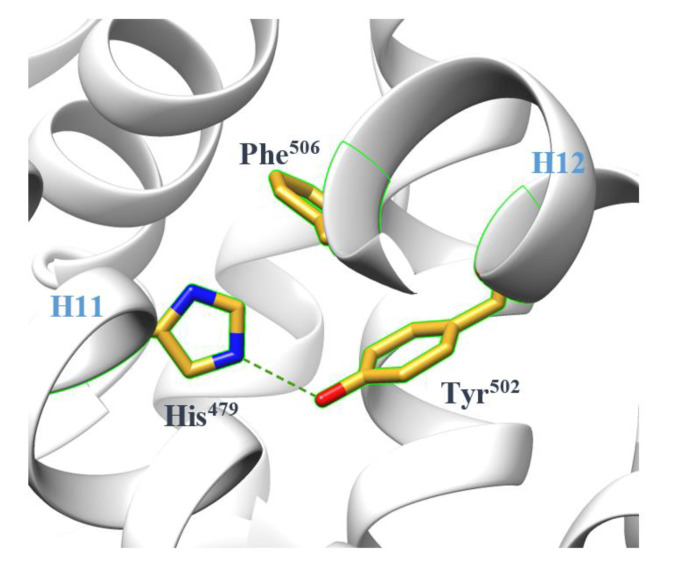
The agonist lock stabilizes H12 through a hydrogen bond (green dashed line) in the agonist-bound RORγ LBD (PDB 4S14).

**Figure 3 ijms-21-05329-f003:**
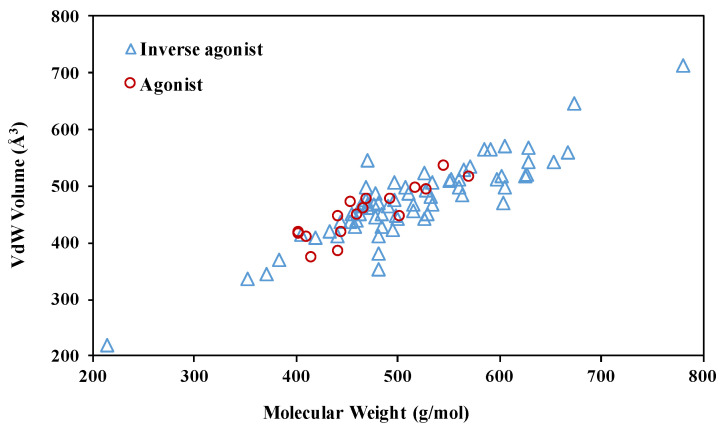
Physical features of the RORγ orthosteric modulators, including 18 agonists (Appendix A) and 69 inverse agonists (Appendix A). For every compound, the Van der Waals (VdW) volume of its binding pose in the orthosteric pocket was measured with YASARA [30].

**Figure 4 ijms-21-05329-f004:**
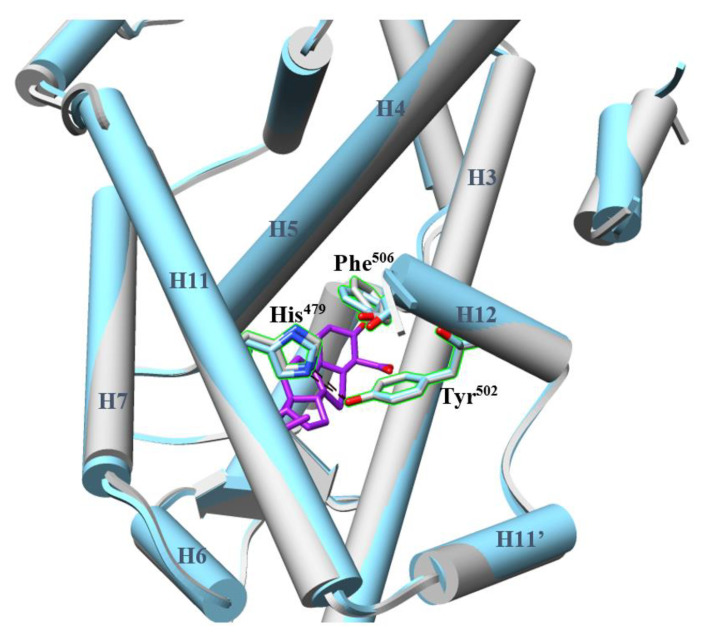
The conformational change of RORγ LBD before and after agonist binding in the orthosteric pocket is minimal. The LBD (PDB 4S14, cyan) bound with the agonist (purple) is superimposed against the apo LBD (grey). The backbones of the LBD are presented in cylinder shape, while the agonist and the residues His^479^, Tyr^502^ and Phe^506^ are in stick shape. The agonist retains the architecture of the pocket and the hydrogen bond of the agonist lock (shown as black dashed lines).

**Figure 5 ijms-21-05329-f005:**
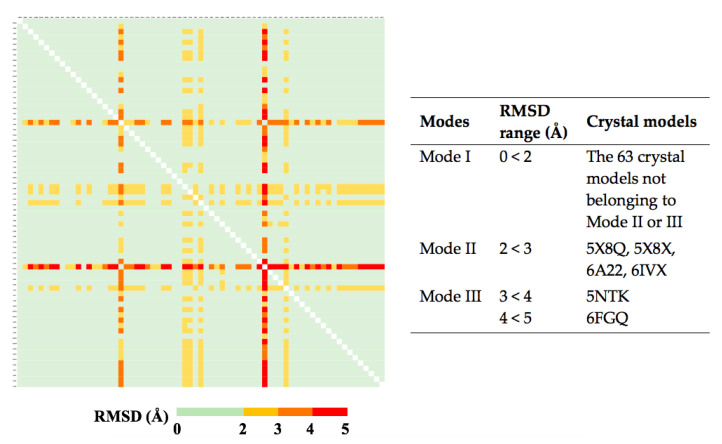
Conformational comparison of RORγ orthosteric pockets induced by 69 inverse agonists. For each ligand-bound LBD, the H1–H11 (265–490 a.a.) portion was extracted for pairwise comparison. The root-mean-square deviation (RMSD) value between two models was measured in the Cα manner by YASARA and then labeled on the figure based on the color scale. According to their RMSD values, the conformations of the pockets were assigned into three modes, which are briefly listed in the right table. The detailed list can be found in Appendix A.

**Figure 6 ijms-21-05329-f006:**
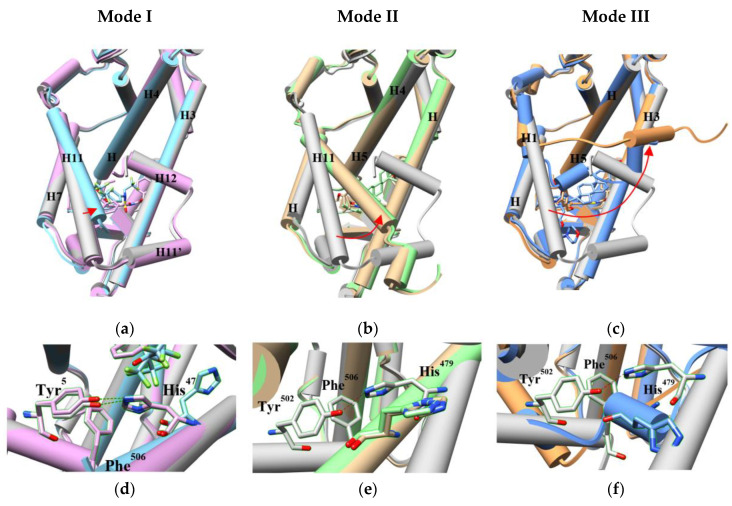
Three different conformation modes of the RORγ orthosteric pocket (**a**–**c**) and the changes of the agonist lock (**d**–**f**) induced by orthosteric inverse agonists. The apo LBD model is shown in grey. The H11 movement is demonstrated by red arrows. (**a**,**d**) Two crystal models in mode I with bound ligands are presented in cyan (PDB 4NB6) and plum (PDB 4NIE). (**b**,**e**) Two crystal models in mode II with bound ligands are presented in light green (PDB 5X8Q) and tan (PDB 6A22). (**c**,**f**) Two crystal models in mode III with bound ligands are presented in blue (PDB 5NTK) and light brown (PDB 6FGQ).

**Figure 7 ijms-21-05329-f007:**
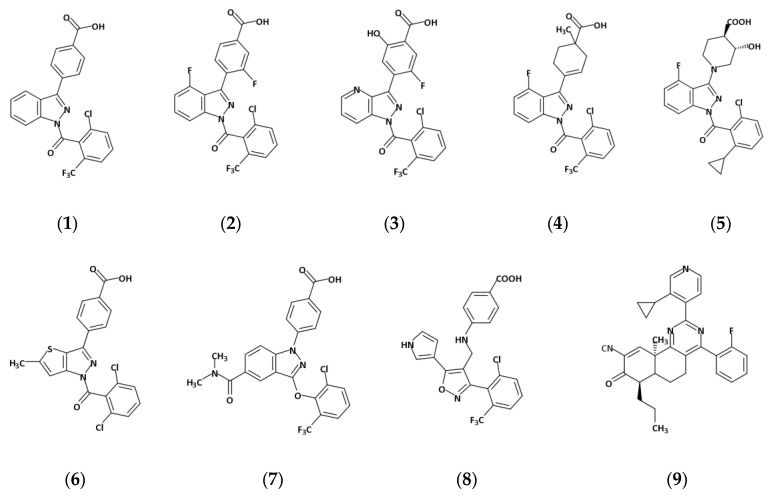
Allosteric inverse agonists binding in the AF-2 site. The corresponding PDB codes for the co-crystal models of the compounds binding in the RORγ LBD are listed as follows: (**1**) 4YPQ, 5C4O [11]; (**2**) 5C4S [11]; (**3**) 5C4U [11]; (**4**) 5C4T [11]; (**5**) 6UCG [37]; (**6**) 6TLM [38]; (**7**) 5LWP [39]; (**8**) 6SAL [40]. No PDB code is available for compound (**9**) [41].

**Figure 8 ijms-21-05329-f008:**
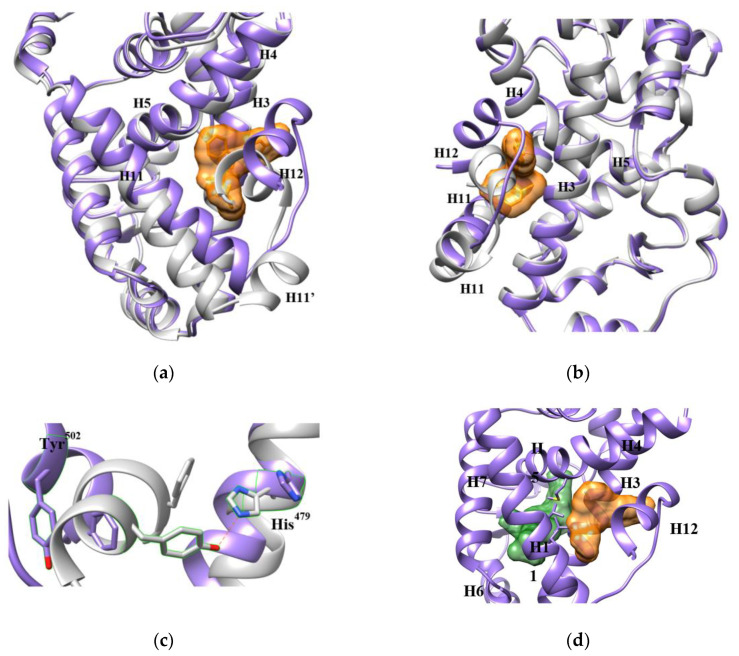
An allosteric inverse agonist **1** binds in the AF-2 binding pocket. The structure of RORγ LBD bound with **1** (orange surface) is presented in purple (PDB 4YPQ), while the apo LBD is in grey. (**a**,**b**) Two views of the pocket from different orientations display the movement of the helices. Especially, **1** partially occupies the space of H12 and forces it to shift away from other helices. (**c**) Affected by **1**, Tyr^502^ is pushed to a position too far away from His^479^ to maintain the agonist lock. (**d**) The AF-2 site is adjacent to the orthosteric site (green) with a short tunnel connecting them, which is surrounded by hydrophobic residues shown in sticks. The orthosteric site is demonstrated with an orthosteric inverse agonist from PDB 5UFO.

**Figure 9 ijms-21-05329-f009:**
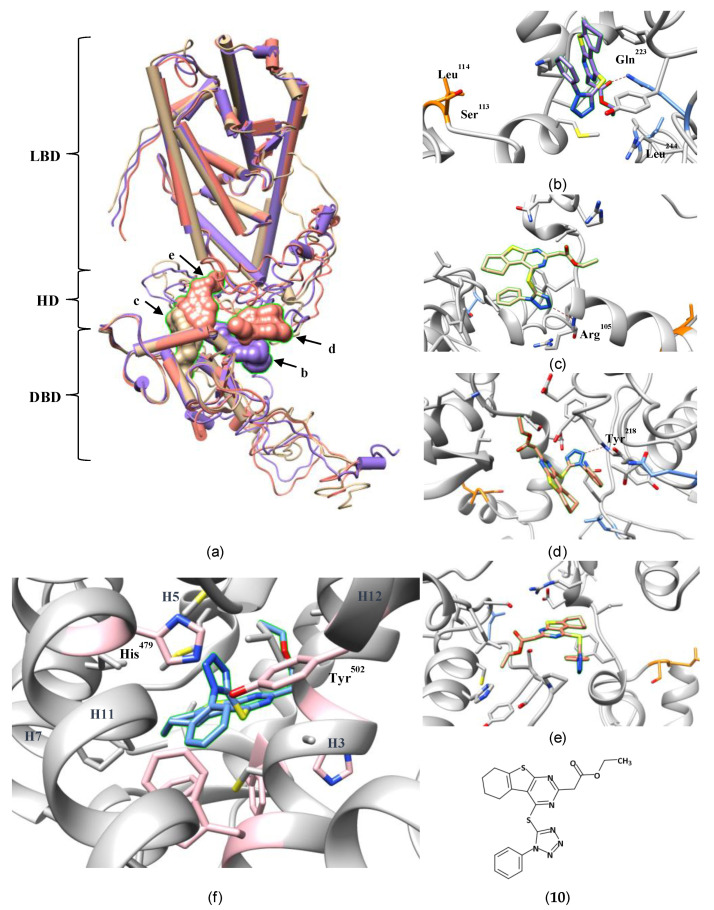
An inverse agonist (**10**) potentially binds in an allosteric pocket of RORγ hinge domain (HD). The details of the full-length RORγ homology models and molecular docking experiment were described in Supplementary Protocol S1. (**a**) The top-ranked poses of **10** docked against the HD site in three different homology models are displayed in purple, tan and salmon colors, respectively. The binding energy of the poses in the homology models was analyzed (Appendix A). (**b**) In the purple model, **10** poses at a position closed to Lao’s. It hydrogen-bonds to Gln^223^ but stays far from Leu^244^. Ser^113^ and Leu^114^ are presented in orange color and Gln^223^ and Leu^244^ in light blue. (**c**) In the tan model, **10** shows no contact to Ser^113^, Leu^114^, Gln^223^ or Leu^244^. (**c**,**d**) In the salmon model, two poses of **10** were docked in the site. None of them is closed to Ser^113^, Leu^114^, Gln^223^ or Leu^244^. (**f**) **10** may bind in the orthosteric pocket of RORγ LBD (PDB 4NIE) through π-stacking (pink) and hydrophobic (grey) interactions. It may contact the agonist lock. More details are shown in Appendix A.

**Figure 10 ijms-21-05329-f010:**
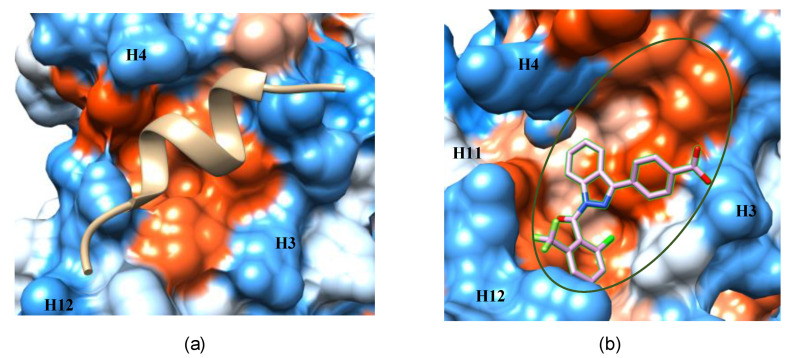
The coactivator-binding pocket ((**a**), PDB 3KYT) and the extended AF-2 binding pocket ((**b**), PDB 4YPQ). The structure models are presented by hydrophobic surface with red color for high level of hydrophobicity, blue color for high level of hydrophilicity. The extended AF-2 pocket is circled by green line.

**Figure 11 ijms-21-05329-f011:**
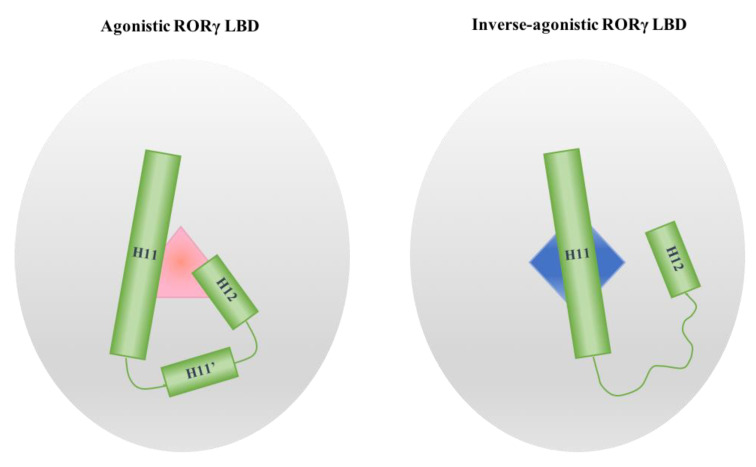
Schematic presentations of the conformational difference of H11, H11′ and H12 in RORγ LBD bound with an agonist (pink triangle) and an inverse agonist (blue diamond).

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
