# Peer review of "RORγ Structural Plasticity and Druggability"

_ijms, 2020, doi:10.3390/ijms21155329_

Round 1

Reviewer 1 Report

Overall, well written manuscript and found scientific significance but some points to be discussed for better understanding of the manuscript which are given below.

The current manuscript includes recent literature which will be given better understanding of the disease to the researcher in Nonalcoholic Fatty Liver Disease and Colorectal Cancer. The immunopathogenesis of Nonalcoholic Fatty Liver Disease and Colorectal Cancer: is complex and involves innate and adaptive immune responses, including many adipokines, chemokines and immune mediators, and not only leptin and adiponectin. Thus, the authors should also discuss other adipokines and chemokines which have a significant role in the immunopathogenesis of disease.  

Author Response

Overall, well written manuscript and found scientific significance but some points to be discussed for
better understanding of the manuscript which are given below.
The current manuscript includes recent literature which will be given better understanding of the
disease to the researcher in Nonalcoholic Fatty Liver Disease and Colorectal Cancer. The
immunopathogenesis of Nonalcoholic Fatty Liver Disease and Colorectal Cancer: is complex and involves
innate and adaptive immune responses, including many adipokines, chemokines and immune
mediators, and not only leptin and adiponectin. Thus, the authors should also discuss other adipokines
and chemokines which have a significant role in the immunopathogenesis of disease.

Authors: We thank the reviewer for helpfully suggesting additional discussion of Th17-related diseases.
We added additional information about relevant diseases in our first paragraph.

Reviewer 2 Report

The article by Huang et al. presents an overview of the current knowledge of the ROR gamma structural properties and druggability. This trascription factor is implicated in various biological processes and thus becomes a relevant drug target. In this paper a very interesting work correlating the plasticity of two different binding sites (orthosteric and allosteric), their druggability, and the nature of the ligands is presented.

The manuscript is well written and presents an interesting work comparing and synthesizing a lot of structural data obtained experimentally through X-ray cristollography or theoretically through homology modeling, molecular docking or molecular dynamics.

However, in the following sections I make comments and/or raise questions that need to be taken into account and answered in order to improve the manuscript and make it suitable for publication.

As a general comment, it is clear that this review gather a huge work of examining the different LBD crystal structures available in the PDB. I suggest, for each of the figures presented in the present manuscript that the authors indicate clearly the PDB IDs that were used to render the snapshots and illustrate their different hypothesis and/or conclusions.

* On line 68 and following, while commenting on Fig. 1, alpha-helices H3, H5, H7 and H11 are mentioned but the are not identified on the figure. This can be misleading for the reader. The identification of these helices would help in the recognition of the pocket that is evoked on line 68.

* On line 97: "In the agonist-bound LBD, a hydrogen bond formed between His 479 and Tyr 502 locks H12 in the correct position under the assistance of Phe 506"

In ref 22 which is cited in reference to the agonist lock, there is no mention of Phe 506 residue, and from the representation as depicted in Fig. 2, it is difficult to see how this residue Phe 506 contributes to the stabilization of the hydrogen bond or H12 helix. How do the authors come to this conclusion ?

* Figure 2

To improve understanding and readability, I suggest to: (i) identify/label H12 helix, (ii) indicate the pdb structure used to create the figure.

* Figure 3

For clarity reasons, the units for the Molecular Weight (x-axis) should be specified.

I can not see any difference between Inverse agonist and Agonist since the symbol for Inverse agonist is not visible.

How many Agonists and Inverse agonists were considered to create the graph ?

Some explanation about the evaluation of the volume of the Agonists and Inverse agonists would be welcome.

* On line 111, there is a mention of “18 co-crystals of LBD-agonist complexes”

I would suggest to give (in supplementary data for instance) the PDB ID of the 18 different crystal structures from which the Calpha RMSDs were evaluated.

* On line 113: "The modulators interact non-covalently with H3, H5, H6,and H7 to stabilize the whole pocket and hydrophobically contact His 479 while remaining distant from Tyr 502 (Fig. 4)."

In Fig. 4, is a modulator represented ? If yes, which color is used ? H6 is not labeled whereas it is mentioned in the text. These missing elements make it hard for the reader to correlate the text and the figure. Which drawing and coloring methods are used for Tyr 502 and His 549 ?

The short legend does not specify this and it is quite hard to correlates the representation with the text.

* On line 121-122: the same way the authors indicate the number of co-crystals of LBD-agonists complexes used for the computation of backbone Calpha RMSDs (line 111), I suggest they also indicate the number of co-crystals of LBD-inverse agonist complexes that are used for the computation of pairwise backbone Calpha RMSD and for the rendering of Fig. 5 (from the figure, I would say 69, but ...). As previously suggested, the ensemble of PDB IDs used in the context of the writing of the review could be listed in supplementary data.

* Figure 5

One would expect a square to represent the 69 x 69 symmetric matrix.

The PDB IDs are not readable, and maybe the addition of a table listing all the structures as well as their belonging to each Mode (I, II or III) would be insightful.

* On line 125: "The inverse agonists in this group pushed or pulled H3, H4, H5, and/or H7 to slightly deviate from the original positions (Fig. 6A)"

There is no label corresponding to H3, H4, H5 and/or H7 in Fig. 6A. One can clearly see some deviation between H11 from the different models, but it is not mentioned in the text.

* On line 127: "Although H11 retains the backbone pose, the side chain of His 479 may flip (Fig. 6D), which breaks the agonist lock [26]".

There must be an error in the citation because I could not find any mention of the agonist lock, break of the agonist lock or any residues implicated in the formation of this one in the reference 26 (Structure-based design of substituted hexafluoroisopropanol-aryl-sulfonamides as modulators of RORc, Fauber et al)

* On line 128: "the side chains of Lys336 and Glu504"

It would be informative to indicate in the legend of Fig. 6 if these residues are represented and to label them on the figure or to indicate the color in which they are represented.

* On line 132: for clarity reasons, the same way RSMD values were associated to Mode I and Mode II, values of the RMSD associated to Mode III should be specified. One could think that Mode III correspond to color orange and red from Fig. 5. Is it really the case ?

* On line 133: there is a typo; His479 should replace His497.

* Figure 6

I suggest to write a legend containing substantial information answering the following questions/points:

What are the PDB IDs used to elaborate the different panels of the figure ?

Labels for the residues mentioned in the text are missing

It is difficult to know if some ligands are represented in panels A, B and C. We could have the impression that small molecules (compared to LBD) are present. But neither the legend, nor the text, gives any clear information about that. From the text, we could think that the ligand is represented.

For panels A, B and C, it would make sense to use exactly the same orientation of the protein: it would be easier to compare Modes I, II and III.

* Figure 7

On line 155 of the manuscript, only four inverse agonists are evoqued in the text (1, 2, 3, 4): in that case is it relevant to have the representation of 9 molecules. Would it be possible to give their name (in the legend for instance)? It would be interesting to know to which reference and PDB IDs there are linked.

* On line 156: "This site is a shallow pocket nested by H3, H4, H5, H11, lidded by H11’ and H12, and mainly filled with hydrophobic residues (Fig. 8)"

Once more, not all the elements mentioned in the text are seen on Fig. 8. Where are H3 and H5 ? If it is not possible to see them because they are in the back/behind other elements, it might be worth adding another panel with a different orientation allowing to see these helices.

* Figure 8

The legend should indicate the meaning of the different colors and the PDB IDs they correspond to. I would suggest to add a panel illustrating the position of the two binding sites: the orthosteric one and the allosteric one. It should be indicated in the legend that the ligand that is in the pocket in the first one from Fig. 7.

* Figure 9

The legend should indicate the meaning of the different colors and the PDB IDs they correspond to.

* On line 165: "We compared the binding modes of nine ligands. Eight ligands were co-crystallized with the LBD [7,32–35]. A ninth was modeled from MD simulations [36]"

Are the nine ligands the ones presented in Fig. 8 ? In that case, maybe it should be indicated for clarity reason.

* On line 170: "First, the pocket is surprisingly pliable. It is smaller in the unbound state until a ligand induces expansion"

It would seem reasonable to give some estimation of the volumes in order to support this statement by quantitative data.

* On line 175: "Second, the AF-2 pocket is connected to the LBD orthosteric pocket through a short tunnel, that can accommodate parts of some ligands"

In the same way of the previous comment concerning Fig. 8, some illustration showing the position of each pocket (orthosteric and allosteric respectively) and the tunnel that exist between the two would be very useful (maybe some adaptation of panel C of Fig. 3 of ref 36).

* On line 195: "The SAR of the most active molecules (10) from the screen …"

Does “(10)” refers to the number of the most active molecules or to the identity of a single molecule ?

* On line 204: It would be interesting and insightful to give some information about the quality of the homology model that was obtained by the authors in order to reproduce the docking experiments of Lao et al.

* On line 204: "One of the top-ranked binding poses occupies a position close to Lao’s but interacts with the surrounding residues differently (Fig. 10B-E)"

Could the author specify which binding pose is close to the one obtain to Lao's ?

* Figure 10

panel A → do the three colors used for the cartoon representation correspond to three different models obtained from I-TASSER ? This information should be given, and it is relevant to comment somewhere about the differences of the various models. Did the authors calculate some RMSD between the models in order to quantify their discrepancies?

Since there are 4 panels B to E, one could think that four ROR models from I-TASSER are used for docking, but in panel A, there are 3 different visible color and thus we could think that there are 3 models generated from I-TASSER. Could the authors clarify this point ?

Is this the same model of ROR that is used for panels B to E ? From the text, one could think that a single model is used for docking, but from the legend of the figure it is not clear.

panel B → it seems that there are more Ser113 labels than needed ....

* General comment about docking experiments:

It would be more rigorous to specify the docking software that was used (if it is the same as Lao et al. then AutodockVina ?), to give the parameters of the experiments and also the number of independent experiments that were performed. Also it is of great interest to compare the free energy of binding issued from the experiments. I strongly recommend to include these data and discuss them in the manuscript.

* Figure 11

I would recommend to indicate H3, H4 and H12 on the snapshot in order for any reader to connect what he sees on the figure with the description given on line 231 of the manuscript.

* Figure 12

There is a mention in the text of a junction between the hydrphobic pocket of the coactivator peptide and the AF-2 allosteric site (lines 233 and 234) → in order to visually connect the two, maybe some helix labels should be added in the figure.

* On line 291 --> "an MD simulation on the complex of 10 and the full-length RORγ homology model (Fig. 10B) will provide a view"

Does this mean that the MD simulations is setup and ready to run ?

* Replace helixes by helices throughout the manuscript

References

The authors should choose between these two ways of writing:

Bioorg. Med. Chem. Lett. OR Bioorganic & Medicinal Chemistry Letters

This way the name of the journal will be the same for references 5, 22, 24, 26, 27, 33 and 36.

* ref 19: the name of the journal is Genes to Cell and not Genes Cells

* ref 25: the article number is missing (17249)

* refs 44 and 51: include the name of the journal and revise the listing of authors (as for the other references, mention the last names of the authors)

* ref 48: add issue and pages of the journal (34, 201–217)

* On line 42 [5][6] should be replaced by [5,6] in order to be homogeneous in the way of citing the references.

Author Response

Thank you for the detailed, helpful comments. Please see the attachment.

Round 2

Reviewer 2 Report

The suggested modifications were integrated in the revised version. The authors have done a very good job in improving the figures and their captions.